# PTEN Protein Phosphatase Activity Is Not Required for Tumour Suppression in the Mouse Prostate

**DOI:** 10.3390/biom12101511

**Published:** 2022-10-19

**Authors:** Helen M. Wise, Adam Harris, Nisha Kriplani, Adam Schofield, Helen Caldwell, Mark J. Arends, Ian M. Overton, Nick R. Leslie

**Affiliations:** 1Institute of Biological Chemistry, Biophysics and Bioengineering, Riccarton Campus, Heriot Watt University, Nasmyth Building, Edinburgh EH14 4AS, UK; 2Patrick G Johnston Centre for Cancer Research, School of Medicine, Dentistry and Biomedical Sciences, Queen’s University, 97 Lisburn Road, Belfast BT9 7AE, UK; 3Edinburgh Pathology, Cancer Research UK Edinburgh Centre, University of Edinburgh, Western General Hospital, Crewe Road South, Edinburgh EH4 2XR, UK

**Keywords:** tumour suppressor, prostate cancer, PTEN, phosphatase, PI 3-Kinase

## Abstract

Loss PTEN function is one of the most common events driving aggressive prostate cancers and biochemically, PTEN is a lipid phosphatase which opposes the activation of the oncogenic PI3K-AKT signalling network. However, PTEN also has additional potential mechanisms of action, including protein phosphatase activity. Using a mutant enzyme, PTEN Y138L, which selectively lacks protein phosphatase activity, we characterised genetically modified mice lacking either the full function of PTEN in the prostate gland or only lacking protein phosphatase activity. The phenotypes of mice carrying a single allele of either wild-type *Pten* or *Pten^Y138L^* in the prostate were similar, with common prostatic intraepithelial neoplasia (PIN) and similar gene expression profiles. However, the latter group, lacking PTEN protein phosphatase activity additionally showed lymphocyte infiltration around PIN and an increased immune cell gene expression signature. Prostate adenocarcinoma, elevated proliferation and AKT activation were only frequently observed when PTEN was fully deleted. We also identify a common gene expression signature of PTEN loss conserved in other studies (including *Nkx3.1, Tnf* and *Cd44*). We provide further insight into tumour development in the prostate driven by loss of PTEN function and show that PTEN protein phosphatase activity is not required for tumour suppression.

## 1. Introduction

Loss of function of the PTEN tumour suppressor is among the most frequently observed genetic events that drive prostate cancer [1,2]. The most common mechanism of loss is deletion of a single *PTEN* gene copy, which is usually associated with loss of PTEN protein expression [2]. Notably, the occurrence of PTEN loss is higher in metastatic disease (seen in 40–60% of these cases) than in primary tumours (10–40%) [3,4,5] and associates with poor prognosis [6,7,8,9,10]. It has therefore been proposed that PTEN status could be used to distinguish between indolent and progressive prostate cancer and accordingly, PTEN is a component of several biomarker signatures which appear to have some power to identify aggressive disease [11,12,13,14,15,16,17,18,19].

PTEN is a core component of the class I phosphoinositide 3-Kinase (PI3K) signalling network, acting as a lipid phosphatase to dephosphorylate the PI3K products, phosphatidylinositol 3,4,5-trisphosphate (PIP_3_) and probably also phosphatidylinositol 3,4-bisphosphate (PI(3,4)P_2_) [20,21]. PI3K and its product lipids play conserved roles regulating metabolism, and promoting the growth, proliferation and survival of many cell types and additionally influencing cell polarity in a more lineage specific manner [21,22,23]. These downstream consequences of PI3K activity are mediated by a large diverse group of direct phosphoinositide lipid-binding proteins, amongst which a dominant role is played in many processes by the AKT group of protein serine/threonine kinases [23,24,25]. Accordingly, there have been intense efforts to develop small molecule inhibitors of PI3K, AKT and the downstream kinase mTOR as drugs to treat cancers including prostate cancer. The success of these agents in clinical trials has been disappointing [26], but notably the AKT inhibitor Ipatasertib has recently been shown to increase progression free survival in a phase III trial in prostate cancer patients displaying PTEN loss [27].

Heterozygous *Pten*^+/−^ mice succumb to a broad range of tumours [28] and genetic deletion of *Pten* specifically from the developed prostate gland causes rapid high grade prostatic intraepithelial neoplasia (PIN) and later invasive prostate carcinoma, with kinetics which seem to depend on genetic background [29,30]. While there is compelling evidence that the lipid phosphatase activity of PTEN, functionally opposing PI3K, is its dominant mechanism of tumour suppression [31,32,33], PTEN also has other potential mechanisms which may contribute [34,35]. These include phosphatase-independent functions in the nucleus and elsewhere [35,36]. PTEN also has weak but robust phosphatase activity in vitro against protein and phosphopeptide substrates, with highest activity against acidic phosphotyrosine substrates [37], and a number of roles and potential substrates for this protein phosphatase activity have been proposed [38,39,40,41,42,43,44,45]. However, confident determination of the substrates of protein phosphatases is challenging and to date a clear picture is yet to emerge regarding the significance of these proposed substrates in PTEN function.

To test the significance of the protein phosphatase activity of PTEN, we have previously engineered a PTEN mutant, PTEN Y138L, which retains activity against lipid substrates yet lacks the normal activity of PTEN against phosphopeptides [46]. PTEN Y138L retains the ability to suppress AKT phosphorylation in cultured cells yet unlike the wild-type enzyme it fails to inhibit glioma cell invasion [47] or control epithelial 3D lumen formation [48]. Notable in these studies, the activity of PTEN Y138L in these cell-based assays could be rescued by mutation of Thr366 but did require lipid phosphatase activity. This indicates that, at least in these assays, the only requirement for the protein phosphatase activity of PTEN is the autodephosphorylation of this residue in the PTEN C-terminus [47,48]. Here, we have generated organ-specific knock-in mice expressing PTEN Y138L to test the requirement for the protein phosphatase activity of PTEN for tumour suppression in the prostate.

## 2. Materials and Methods

**Mice**: The constitutive endogenous gene knock-in *Pten^Y138L^* mouse line was developed as a service by Taconic-Artemis (Cologne, Germany) by homologous recombination in C57BL/6 NTac ES cells and has been studied for spontaneous tumour formation in heterozygosity (Priyanka Tibarewal, Laura Spinelli and Nick Leslie unpublished data). These mice had been back-crossed to C57BL/6J mice for at least 10 generations. The conditional *Pten* allele [49,50] and the prostate-specific *PB4-Cre* [51] mouse lines have been previously described. The *PB4-Cre* line was provided by the NCI Frederick Mouse Repository (Frederick, MD, USA). During breeding, the *PB4-Cre* allele was only allowed to pass through male germline [52]. Mice were maintained in randomly assigned cages in the University of Edinburgh Biological Research Facility following all facility guidelines. All experiments were approved by the University of Edinburgh Ethical Review Committee and performed in accordance with the UK Animals (Scientific Procedures) Act 1986 and following UK Home Office guidance. Mouse PCR genotyping used template DNA released from ear punches with Microzone microLYSIS reagent (Microzone, Stourbridge, UK) and Kapa TAQ (Merck/Sigma, Darmstadt, Germany) following manufacturers protocols. The genotyping primers used were: CreF—CCATCTGCCACCAGCCAG; CreR—TCGCCATCTTCCAGCAGG; PtenY138LF—ATGGAAAGGAGTAAATGGATGG; PtenY138LR—GGAGTAAAAGCAGGAGAATTGG; PtenFloxF—GCCTTACCTAGTAAAGCAAG; PtenFloxR—GGCAAAGAATCTTGGTGTTAC.

To compare spontaneous tumorigenesis, mice were maintained until tumour formation was evident by palpation, other signs of ill health or 600 days of age in tumour-free animals. At sacrifice, the prostate, pelvic lymph nodes and visceral organs were fixed for histology and IHC. Any large tumours were divided for both formalin fixation and immediate appropriate lysis for RNA and protein analysis.

**Antibodies and Immunohistochemistry**: The antibodies used for Immunohistochemistry (IHC) were: Phospho-S473 AKT Antibody #9271 from Cell Signaling Technology (also used for immunoblotting); PTEN antibody Clone 138G6 #9559 from Cell Signalling Technology; Androgen Receptor (AR) from DAKO; PKM2 antibody from R&D systems (AF7244); RAC1-GTP mAb from NewEastBio #26903.

IHC followed Leica BOND Protocols as follows. Androgen Receptor IHC used Leica’s defined ‘Mouse IHC’ protocol; epitope retrieval (ER) was achieved using solution ER1 for 20 min and a 1/100 primary antibody dilution was used. IHC for Ki67 used the ‘Mouse IHC’ protocol with an extended 1h antibody incubation, ER1 for 20 min and undiluted primary antibody. RAC1 IHC used the ‘Mouse on Mouse’ protocol, ER1 for 20 min and a 1/100 primary antibody dilution. P-473AKT IHC followed the ‘Mouse IHC’ protocol, 20 min ER1 and a 1/50 dilution of primary antibody. Caspase 3 IHN followed the ‘Mouse IHC’ protocol, with 20 min ER2 and a 1/50 primary antibody dilution.

**Cell Culture and Immunoblotting**: LNCaP cells were purchased from ATCC (designated LNCaP clone FGC) and cultured in RPMI-1640 with 10% FBS. Cells were transduced with pHR-SIN lentiviruses encoding human PTEN, cells lysed and protein expression and phosphorylation investigated as previously described [47].

Phospho-S240/244 Ribosomal Protein S6 was from Cell Signaling Technology, clone D68F8, #5364. Immunoblotting (but not IHC) for PTEN used an antibody (clone A2B1) from Santa-Cruz Biotechnology. GAPDH immunoblotting used MAB374 antibody from Merck Millipore.

**Gene expression analysis**: Prostates were dissected from 4 mice of each genotype sacrificed at the ages of 6 weeks and 20 weeks of age. RNA was purified from all pooled lobes using an RNAeasy kit (Qiagen, Hilden Germany) and gene expression analysed by hybridisation to Affymetrix GeneChip Mouse ST2.0microarrays as a service by Cambridge Genomic Services (Cambridge, UK).

**Microarray data processing and functional analysis**: The 6-week samples were processed in a single batch, however the 20 week samples contain data from two different batches and a clear batch effect was observed (Appendix A). We applied ComBat [53] to mitigate batch effects followed by RMA with quantile normalisation. Analysis with ArrayQualityMetrics [54] eliminated two samples from the six week dataset and two samples from the 20 week dataset. Annotation data was obtained using the AnnotationDbi and clariomsmousehttranscriptcluster.db Bioconductor packages [55]. Empirical Bayes and topTable approaches were used in the limma package [56] to calculate ANOVA comparisons between samples and Benjamini-Hochberg corrected *p*-values (*q*-values) [57]. The following contrasts were used to construct the Contrast Matrix for the 6 week and 20 week eBayes linear models: “WW-WF”, ”WW-FF”, ”WW-YF”, ”WF-YF”, ”YF-FF”, ”WF-FF”. We took thresholds of *q* < 0.05 and fold-change > 1.5 to identify differentially expressed genes (DEGs). We carried out Functional Annotation Clustering [58] on the DEGs, taking an enrichment score threshold of 1.3 to identify significant Annotation Clusters and using detected genes to define the reference (background) annotations.

**Comparison across public *Pten* null prostate gene expression data**: Six paired gene expression datasets published and archived in GEO from wild-type and Pten null mouse prostates were identified: GSE25140; GSE56470; GSE96545; GSE76822; GSE24691; GSE98493 [12,30,59,60,61,62]. These were analysed using the GEO2R platform to identify gene lists from each study which vary in their expression between wild-type and Pten null prostates with adjusted *p* < 0.05. Comparison between the lists and our own 6 week and 20-week data then identified sets of genes which were altered in all four young (6–12 weeks) or all four older (15–30 weeks) groups in each case.

## 3. Results

To confirm the ability of PTEN Y138L, a mutant which selectively lacks protein phosphatase activity, to regulate AKT in prostate cells in the manner of wild-type PTEN, it was expressed transiently in unselected LNCaP cells using lentiviral vectors. These prostate cancer cells lack PTEN and display elevated phosphorylation of AKT and S6 which is reduced upon expression either of wild-type PTEN or PTEN Y138L but not mutants lacking lipid phosphatase activity, PTEN C124S or PTEN G129E (Figure 1A and Appendix A).

To generate mice with this genetic modification specifically in the prostate gland, we used mice expressing the recombinase Cre driven by the prostate-specific PB4 probasin promoter [51]. These were bred with mice carrying a *Pten* allele flanked by loxP sites [50] and *Pten^+^*^/*Y138L*^ mice expressing the PTEN Y138L mutant from the endogenous *Pten* locus. Notably, the embryonic lethality observed in homozygous *Pten^Y138L^*^/*Y138L*^ mice limits the opportunity to study this homozygous genotype (Priyanka Tibarewal manuscript in preparation). Four *PB4-Cre* positive *Pten* genotypes were used experimentally in our study: *Pten^+^*^/*+*^, *Pten^+^*^/*flox*^*,*
*Pten^Y138L^*^/*flox*^ and *Pten^flox^*^/*flox*^ (Figure 1B,C). 26 mice of each genotype were studied. Cohorts of 6 mice were sacrificed at 6 weeks of age and at 20 weeks for histology, RNA and protein analysis. Further groups of 14 mice were monitored up until the age of 85 weeks (Figure 1D).

In agreement with previous studies [29,30], PIN had formed in *Pten^fl^*^/*fl*^ prostates by 6 weeks of age and later, regions of locally invasive adenocarcinoma in each of these mice by 20 weeks of age (Figure 2A,B and Appendix A). In contrast, in prostates carrying a functional *Pten* gene with lipid phosphatase activity, in either *Pten^Y138L^*^/*flox*^ or *Pten^+^*^/*flox*^ mice we observed a much slower development of disease. All of these mice had foci of PIN by 20 weeks of age, but almost all mice had not developed evident carcinoma by this time (Figure 2A,B). In accordance with this apparent pathology, immunohistochemistry (IHC) showed increased proliferation, revealed by Ki67 staining, and elevated detection of activated phosphorylated AKT phosphorylation and active GTP loaded RAC1 in prostates lacking PTEN (*Pten^flox^*^/*flox*^) but not in those expressing either the wild-type enzyme or the PTEN Y138L mutant protein (Figure 2A,C and Figure 3A). IHC also revealed strong nuclear staining for the androgen receptor (AR) in the majority of cells in the prostates of 6-week-old *Pten^flox^*^/*flox*^ mice. However, in samples from each of the other genotypes, AR staining was weaker and heterogeneous (Figure 3B).

These data analysing progression from PIN to carcinoma, rates of proliferation and increases in AKT phosphorylation and AR protein levels, suggest that only full deletion of PTEN in *Pten^flox^*^/*flox*^ mice drives these changes. Accordingly, analysis of survival in the mice cohorts found that *Pten^flox^*^/*flox*^ mice showed significantly shorter overall survival, with all but one of this group being sacrificed with prostate tumours by 65 weeks of age (Figure 4A). In these phenotypic data, *Pten^+^*^/*flox*^ and *Pten^Y138L^*^/*flox*^ mice appeared similar, showing only slow development of PIN. This implies that the dominant factor in these observations may be the lipid phosphatase activity of PTEN correlating with the regulation of AKT. However, there were notable differences between the high-grade PIN lesions observed in the *Pten^+^*^/*flox*^ and *Pten^Y138L^*^/*flox*^ mice as only the latter cohort showed frequent large regions of lymphocyte infiltration and of more fibroblastic stroma (Figure 4B–E). Lymphocyte infiltration was observed in PIN in all of the *Pten^Y138L^*^/*flox*^ mice and stromal changes observed in samples from three of the six mice.

To gain deeper insight into the changes occurring before and during the development and progression of PIN in these mice, global gene expression analysis was conducted. RNA was purified from the prostates of 4 mice from each genotype at 6 weeks and 20 weeks of age and analysed using hybridisation microarrays, followed by data quality control, differential expression analysis and functional enrichment analysis. Notably the abundance of *Pten* mRNA in *Pten^flox^*^/*flox*^ prostate tissue was 0.16× that found in *Pten^+^*^/*+*^ mice tissue (*t*-test *p* = 0.0022). The number of genes significantly changed in their expression across the four genotypes reflected the observed pathology (Table 1). At 6 weeks of age, gene expression changes were modest (<200 genes significantly different) and the largest numbers were observed in *Pten^flox^*^/*flox*^ prostates relative to each of the other 3 genotypes. At 20 weeks of age, the greatest number of significant changes (>800 genes upregulated; >500 genes downregulated) were observed between wild-type and *Pten* null prostates, with intermediate numbers of genes being differentially expressed between wild-type tissue and prostates carrying a single allele encoding either wild-type or Y138L mutant PTEN. However, the differences in gene expression between prostate tissues of these two genotypes each with a single lipid phosphatase active *Pten* allele (*Pten^+^*^/*flox*^ and *Pten**^Y138L^*^/*flox*^) was very modest (<100 genes differentially expressed).

To identify functional patterns within the gene expression data, Functional Annotation Clustering was performed, integrating annotation data from multiple databases [58]. The gene expression signatures from the 6-week prostates indicated consistent patterns of functional change within the *Pten* null prostate tissues relative to each of the other three genotypes and associating with the development of PIN (Appendix A). Notably, there were only 5 functional clusters upregulated in the *Pten^Y138L^*^/*flox*^ tissue relative to the *Pten^+^*^/*flox*^ tissue, one of which is designated “Inflammatory response/Immunity” and contains genes associated with both innate and adaptive immune systems. This appears consistent with the immune cell infiltration observed during histology (Appendix A and Figure 4E). This functional assignation is associated with significantly increased expression in the *Pten^Y138L^*^/*flox*^ tissue of eight genes previously linked to immune cell function: Naip5, Thbs1, Elf3, Prkcq, Anxa1, Pglyrp1, Reg3g and Ltf.

The availability of gene expression data from several previous studies provided the opportunity to use these to validate consistent changes in gene expression caused by the full deletion of *Pten* from the murine prostate. Therefore, we compared our data (Appendix A) with those from six published tissue specific knock-out studies which also compared gene expression in *Pten^+^*^/*+*^ and *Pten^−^*^/*−*^ prostates [12,30,59,60,61,62]. We used three studies from young mice (6–12 weeks) and three from older mice (15–30 weeks) which were compared to our data from 6 weeks and 20 weeks, respectively. This identified 43 genes which were differentially expressed between wild-type and Pten-null prostates in all four studies of young 6–12-week-old mice and 82 genes differentially expressed in prostate tissue in all of the four independent datasets from older mice between 15 and 30 weeks of age (Appendix A). Notably, these consistently observed changes in gene expression between wild-type and *Pten* null prostates included orthologs of many genes functionally associated with human prostate cancer, such as *NKX3.1* [63], *TNF* [64], *CD44* [65], *KLF5* [66], *ASNS* [67], *CXCL16* [68], *IL1RN* [69], *LY6A*/*SCA1* [70], *GATA3* [71], *TNS1* [72] and *STAT1* [73], and encoding a number of reported relevant biomarkers including ANXA2 [74], KRT19 [75], SDC1 [76], RHOU [77], SEPT9 [78], ANPEP [79], TSPAN8 [80], COL4A6 [81] and ATF6 [82] (Appendix A).

The signal for a microarray probe (18746) specific for the oncogenic M2 isoform of pyruvate kinase was increased in both *Pten^flox^*^/*flox*^ and *Pten^Y138L^*^/*flox*^ samples relative to *Pten^+^*^/*+*^ and *Pten^+^*^/*flox*^ prostates. To confirm this finding, immunohistochemistry specific for the Pyruvate Kinase M2 (PKM2) isoform showed elevated levels in *Pten^flox^*^/*flox*^ prostate tissue of 6-week-old mice relative to other genotypes, particularly in areas of PIN (Figure 5). A similar pattern was also observed for the Keratin 19 gene (*Krt19*) which was elevated at the mRNA level in both *Pten^flox^*^/*flox*^
*and Pten^Y138L^*^/*flox*^ samples, but with IHC showing increased expression of KRT19 protein only in *Pten^flox^*^/*flox*^ samples especially areas of PIN, relative to other genotypes (Figure 5). Notably, KRT19 has been recently described to protect cancer cells from the immune system, acting in concert with transglutaminase 2 (TGM2) [83] which was also found to be elevated at the mRNA level in *Pten^flox^*^/*flox*^ prostates of 20-week-old mice (Appendix A).

## 4. Discussion

We have bred colonies of mice lacking one or both copies of the *Pten* gene specifically in the prostate as well as mice with a single *Pten* gene copy encoding a mutant enzyme with lipid phosphatase activity but not protein phosphatase activity. A key new conclusion of this work is that adenocarcinoma development is driven by full loss of PTEN function in the prostate in a manner that is not observed when PTEN function is partially reduced or protein phosphatase activity is selectively lost. This correlation provides further support for the functional connection between canonical PI3K-AKT signalling and neoplastic prostate pathology. The observation of PIN in prostates with a single copy of either wild-type *Pten* or *Pten^Y138L^* but adenocarcinoma in *Pten* null prostates is consistent with previous observations of the dose dependency of tumour suppression in *Pten* mutant mice [29,30]. It is also consistent with the observation that PTEN Y138L and the wild-type enzyme display very similar phosphatase activity against PIP_3_ and suppression of AKT in cell-based assays (Refs. [46,47] and Figure 1A).

Accordingly, the apparent lack of significance of the protein phosphatase activity of PTEN in prostate tumour supprssion supports, and provides insight into, existing efforts to treat prostate cancers with small molecule inhibitors of PI3K, AKT and mTOR. The results of tens of clinical trials with inhibitors of PI3K, AKT and mTOR in prostate cancer patients have been disappointing with significant adverse effects and little or no efficacy [26,84]. A possible contributing factor to these failures has remained our poor understanding of the processes driving and sustaining these cancers, and dysregulation of protein substrates of PTEN acting independently of PI3K in cells lacking PTEN has been proposed [38,39,40,41,85]. In contrast, our data show that the protein phosphatase activity of PTEN is not required for tumour suppression in the prostate and provide support for efforts to optimise the clinical use inhibitors of AKT signalling in combination therapy [27]. Our data are in accord with the observed phenotypes of mice expressing the stable PTEN G129E mutant allele lacking lipid phosphatase activity but retaining protein phosphatase activity. These mice display a worse tumour phenotype than mice carrying a full null allele and argue strongly against strong independent tumour suppressor functions for the protein phosphatase activity of PTEN [32,33]. Notably, recent studies of mammary tumour formation in these mice provide a possible explanation for these findings. They identify the glucocorticoid receptor as a functional target of PTEN protein phosphatase activity, but acting in opposition to its lipid phosphatase activity as loss of PTEN protein phosphatase activity promoted cell death [45].

A recognised weakness of our study, resulting from the use of the available constitutive knock-in *Pten^Y138L^* allele, is the expression of this allele throughout the *Pten^Y138L^*^/*flox*^ animals. Although no pathology or phenotype has been identified in *Pten^Y138L^*^/*+*^ animals at the relevant ages, it is possible that the phenotypes we observe in the prostates of *PB4-Cre Pten^Y138^**^L^*^/*flox*^ mice may be influenced by systemic changes or the expression of PTEN Y138L lacking protein phosphatase activity in, for example, immune cells. We also recognise that the prostate pathology observed in *PB4-Cre Pten^flox^*^/*flox*^ mice cannot model the genetic and phenotypic diversity of human prostate cancer. Therefore, although we see no evidence for it, we cannot exclude a role for the protein phosphatase activity of PTEN in contributing to tumour suppression in other subtypes of prostate cancer driven by other mechanisms.

Immunohistochemistry for the Androgen Receptor (AR) in the prostates of 6-week-old mice surprisingly showed an upregulation of AR in *Pten* null animals, with a nuclear localisation in most cells throughout the gland, even in areas which appeared morphologically normal. Previous evidence shows that elevation of PI3K-AKT signalling by mechanisms including the loss of PTEN leads to the inhibition of AR-induced gene expression [60,86] and accordingly our analysis of gene expression shows reduced expression of the androgen induced genes *Gnmt, Nkx3.1* and *Pbsn* in 6 week-old prostate tissue lacking PTEN. Therefore, increased AR protein levels and nuclear localisation in these animals seems unexpected and may indicate further complexity in the interactions between the AR and PI3K regulatory systems. No significant effects on the abundance of the AR transcript itself were observed in any of our data, implicating effects at the level of the AR protein. It is noted that AR should bind and activate the Probasin promoter CRE transgene in these animals, but its presence in all genotypes seems to make this an unlikely explanation for these data.

120 genes were identified which were changed in their expression in *Pten^flox^*^/*flox*^ prostates relative to wild-type in all gene expression studies of either young 6–12-week-old mice or older 15–30 week old mice, but only five genes were identified which were downregulated in all 8 samples across both age brackets. One of these five most consistently regulated genes was *Nkx3.1*, which encodes a key transcriptional regulator and tumour suppressor in prostate cancer. It has been recognised that loss of PTEN reduces NKX3.1 expression, but previous research has focused on regulation at the protein level, rather than effects which reduce *Nkx3.1* mRNA levels [63]. These multiple studies identifying distinct mechanisms by which loss of PTEN also reduces NKX3.1 levels implies the existence of a functionally significant conserved link between the two proteins. In broader terms, these data showing the consistency with which gene expression effects are observed in these genetically modified mouse models of prostate cancer also highlights the very limited understanding of how, and with what effects, these many gene expression changes are driven by loss of the PTEN tumour suppressor.

The gene expression studies showed one of the functional clusters upregulated in the *Pten^Y138L^*^/*flox*^ tissue relative to the *Pten^+^*^/*flox*^ tissue was designated “Inflammatory response/Immunity”, and was consistent with the histologically observed increase in lymphocyte and other immune cell infiltrations around the PIN lesions in these mice, indicating association with loss of PTEN protein phosphatase activity. It seems relevant that pan-immune-inflammation correlates with poor prognosis in CRPC, yet most prostate cancers display low levels of lymphocyte infiltration and respond poorly to immunotherapy [87,88], it seems significant that loss of the protein phosphatase activity of PTEN appears to correlate selectively with immune cell infiltration.

## Figures and Tables

**Figure 1 biomolecules-12-01511-f001:**
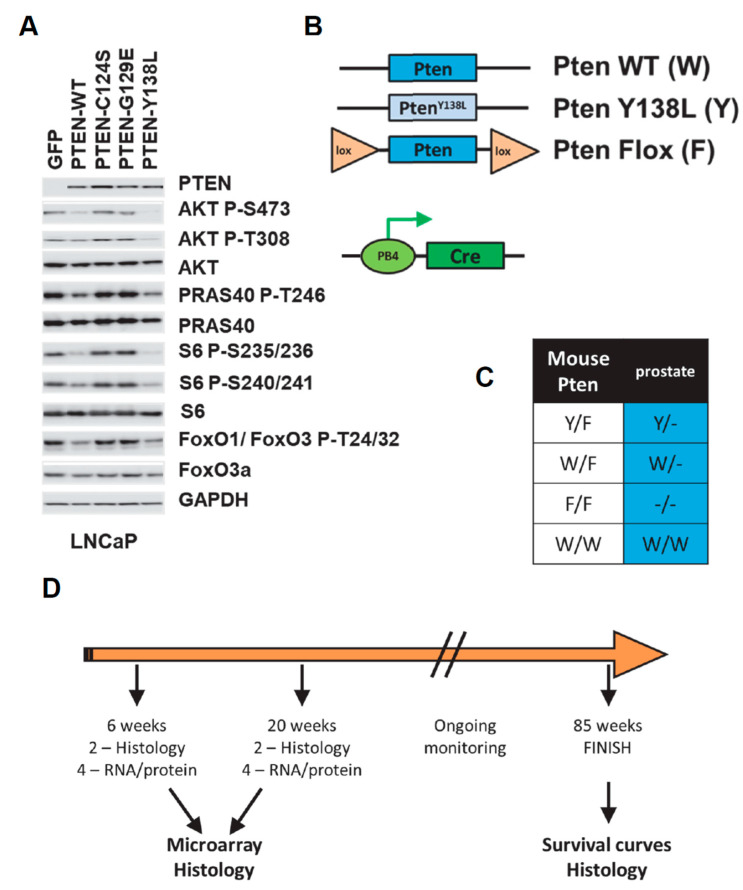
(**A**) Unselected LNCaP cells transiently expressing green fluorescent protein (GFP) or the indicated PTEN proteins were analysed by immunoblotting with the indicated antibodies. Quantitation of this data is shown in Appendix A. (**B**–**D**) The study plan uses 2 mutant *Pten* alleles, a constitutive *Pten^Y138L^* allele and a copy of *Pten* flanked by *loxP* recombination sites (Flox or F) which can be deleted upon expression of the exogenous Cre recombinase. These are represented alongside the wild-type (WT) gene. The PB4-Cre transgene which expresses Cre specifically in the post-natal prostate-epithelium from a modified probasin promoter is also represented. (**C**) The four experimental genotypes used in the study are represented. F represented the undeleted loxP-flanked Pten gene copy which is fully functional until deleted by Cre specifically during prostate development. (**D**) Groups of mice were sacrificed for tissue at 6 weeks and 20 weeks of age whilst a further cohort were monitored for tumour formation up to the age of 65 weeks.

**Figure 2 biomolecules-12-01511-f002:**
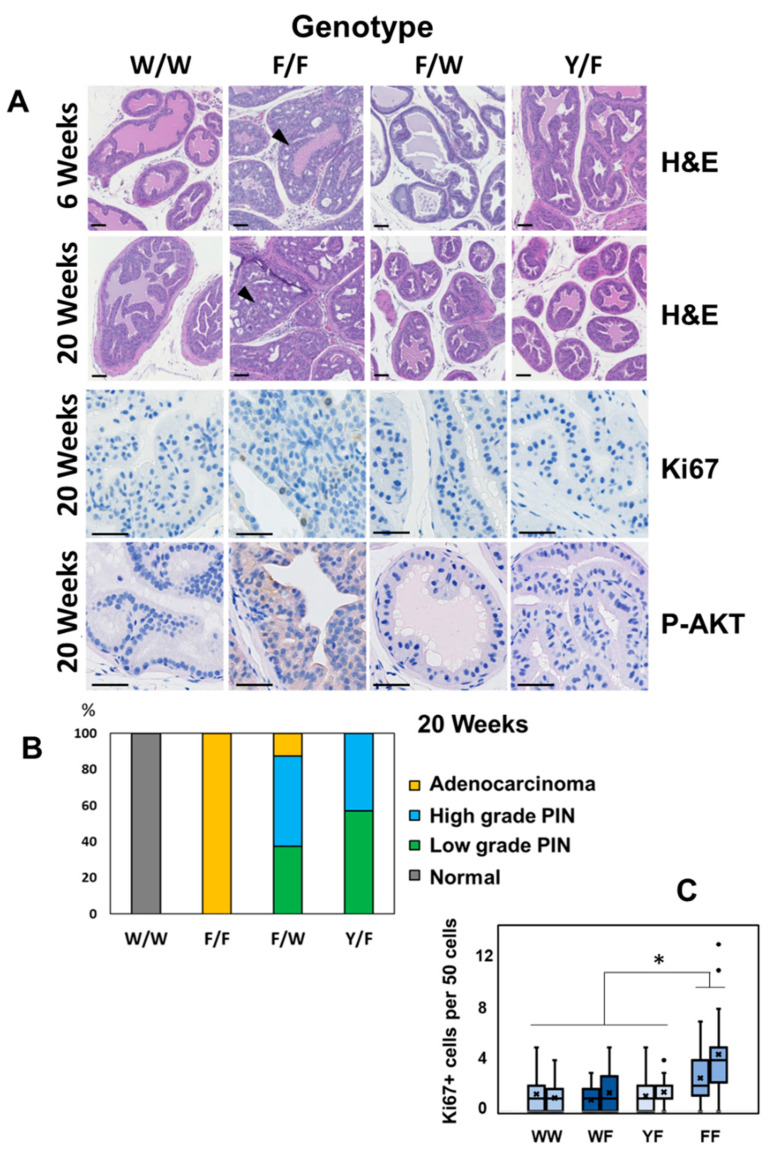
(**A**) Representative sections of H&E (hematoxylin and eosin) stained prostate tissues from the indicated ages and genotypes of mice. Arrowheads show Prostate Intraepithelial Neoplasia (PIN) in 6-week-old Pten F/F mice and adenocarcinoma in 20 week F/F mice. Ki67 immunohistochemistry was used to detect proliferating cells in 20-week sections and parallel samples were also probed by IHC for AKT phosphorylation (Ser-473). Scale bars: 100 μm. (**B**) The most aggressive pathology evident in each prostate at 20 weeks of age is noted for each genotype. N = 6 per genotype. (**C**) The number of Ki67 positive nuclei per 50 cells was counted for 25 regions of the prostate. Data are shown for pairs of mice from the indicated genotypes. Boxes represent quartiles and the cross (x) represents the mean number of proliferative cells per 50 cell group. Each FF sample had significantly more Ki67 positive nuclei than each WW sample (*p* < 0.05 Mann–Whitney U Test) denoted by *. W/W = *Pten^+^*^/*+*^; W/F = *Pten^+^*^/*flox*^; Y/F = *Pten^Y138L^*^/*flox*^; F/F = *Pten^flox^*^/*flox*^.

**Figure 3 biomolecules-12-01511-f003:**
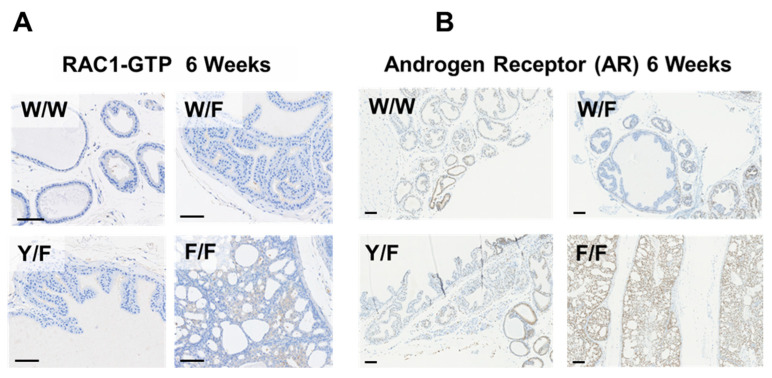
(**A**) Representative RAC1-GTP abundance in sections of prostate tissues from the indicated *Pten* mutant genotypes of mice at 6 weeks of age revealed by immunohistochemistry. (**B**) Representative immunohistochemistry for the Androgen Receptor (AR) in prostates of the indicated genotypes at 6 weeks of age. W/W = *Pten^+^*^/*+*^; W/F = *Pten^+^*^/*flox*^; Y/F = *Pten^Y138L^*^/*flox*^; F/F = *Pten^flox^*^/*flox*^. Scale bars: 100 μm.

**Figure 4 biomolecules-12-01511-f004:**
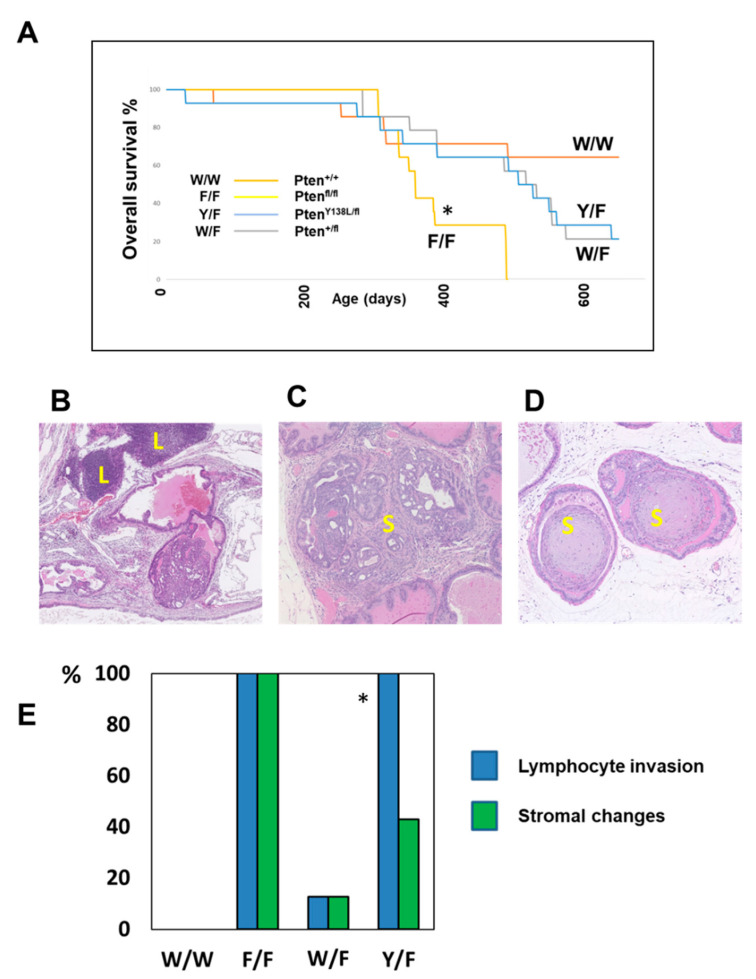
(**A**) Overall survival of cohorts of the indicated *Pten* genotype mice is shown. * Statistical analysis by Log Rank (Mantel Cox) test show a significant difference in survival (*p* < 0.05) between WW vs. FF groups. (**B**–**D**) Sections of H&E (hematoxylin and eosin) stained prostate tissues from F/F mice at 20 weeks of age which illustrate the lymphocyte invasion (L) and stromal changes (S) observed in many of the mice. (**E**) The observation of these changes in each prostate at 20 weeks of age is noted for each genotype. N = 6 per genotype. W/W = *Pten^+^*^/*+*^; W/F = *Pten^+^*^/*flox*^; Y/F = *Pten^Y138L^*^/*flox*^; F/F = *Pten^flox^*^/*flox*^. * 8/8 Y/F mice displayed lymphocyte infiltration, significantly different from the W/F cohort which showed only 1 mouse with lymphocyte infiltration (*p* < 0.005 Fisher’s exact test).

**Figure 5 biomolecules-12-01511-f005:**
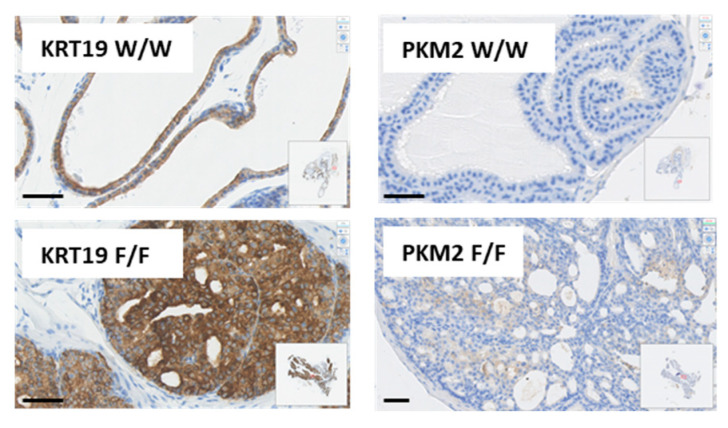
Representative immunohistochemistry showing the abundance of Keratin 19 (KRT19) and Pyruvate Kinase M2 (PKM2) in sections of prostate tissues from wild-type (*W*/*W*) and mice conditionally deleted for Pten in the prostate (F/F) at 6 weeks of age. W/W = *Pten^+^*^/*+*^; F/F = *Pten^flox^*^/*flox*^. Scale bars: 50 μm.

**Table 1 biomolecules-12-01511-t001:** This shows the numbers of genes found to be differentially expressed between mouse prostate samples of the indicated *Pten* genotype pairs. For example, 24 genes were higher in *Pten*^+/Flox^ (W/F) prostates relative to *Pten*^+/+^ (W/W) prostates in samples from 6-week-old mice. Data are shown for these compared genotype pairs from animals at 6 weeks of age (top) and 20 weeks of age (bottom). The inclusion threshold was a difference in log fold change of >1.5 and *p* < 0.05). W/W = *Pten^+^*^/*+*^; W/F = *Pten^+^*^/*flox*^; Y/F = *Pten^Y138L^*^/*flox*^; F/F = *Pten^flox^*^/*flox*^.

**Prostate tissue at 6 weeks of age**						
*Pten* Genotype			Number of genes higher in:			
			W/W	W/F	Y/F	F/F
Relative to:		W/W		24	48	188
		W/F	37		78	191
		Y/F	21	36		186
		F/F	49	54	46	
**Prostate tissue at 20 weeks of age**						
*Pten* Genotype			Number of genes higher in:			
			W/W	W/F	Y/F	F/F
Relative to:	W/W			301	347	865
	W/F		244		90	641
	Y/F		236	61		616
	F/F		518	374	398	

## Data Availability

All data are available on reasonable request from the authors.

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
