# Peer review of "PTEN Protein Phosphatase Activity Is Not Required for Tumour Suppression in the Mouse Prostate"

_biomolecules, 2022, doi:10.3390/biom12101511_

Round 1
Reviewer 1 Report
In this manuscript the authors sought out to clarify the significance of the protein phosphatase activity of PTEN as a well known core component of the PI3K pathway which is widely adopted in several biomarker signatures among different urological malignancies. The authors created an organ-specific knock-in mice expressing PTEN Y138L to test the requirement for the protein phosphatase activity of PTEN for tumor suppression in the prostate.
Overall interesting results. The authors should be commended for such an effort. The manuscript is well written and presented. Methods behind seem to be clearly declared.
From a more clinical perspective, the authors should consider to discuss the research among other urological malignancies in which novel emerging platform of diagnosis, longitudinal monitoring and prognosis was recently reviewed. The stable adoption of - for example - liquid biopsy, novel immunohistochemical assays, seems to be promising across prostate, renal, and urothelial carcinoma as well the response to novel ARTAs, PARP-i or immune check piont inhibitors, or conventional chemotherpy agents. The manuscript would benefit from such parallelism with prostate cancer itself (Actas Urol Esp (Engl Ed). 2020 Apr;44(3):139-147. doi: 10.1016/j.acuro.2019.08.007), (Eur Urol. 2021 Jun;79(6):762-771. doi: 10.1016/j.eururo.2020.12.037), (Tumour Biol. 2022;44(1):107-127. doi: 10.3233/TUB-211568.), (Prostate. 2022 Nov;82(15):1456-1461. doi: 10.1002/pros.24419. Epub 2022 Jul 28.), (N Engl J Med. 2015 Oct 29;373(18):1697-708) (Eur Urol. 2018 Apr;73(4):572-582. doi: 10.1016/j.eururo.2017.10.036. Epub 2017 Nov 10.), (Cancers (Basel). 2021 Sep 8;13(18):4522. doi: 10.3390/cancers13184522), bladder cancer (Eur Urol Oncol. 2021 May 6;S2588-9311(21)00078-X. doi: 10.1016/j.euo.2021.04.004), (Asian J Urol. 2021 Oct;8(4):376-390. doi: 10.1016/j.ajur.2021.05.001), (Eur Urol Oncol. 2021 Apr;4(2):204-214. doi: 10.1016/j.euo.2020.01.003), (Urol Oncol. 2022 Mar;40(3):110.e1-110.e9. doi: 10.1016/j.urolonc.2021.10.010. Epub 2021 Dec 11. ), and renal cell carcinoma (Curr Drug Targets. 2020;21(16):1664-1671. doi: 10.2174/1389450121666200324151056.), (Lancet Oncol. 2022 May;23(5):612-624. doi: 10.1016/S1470-2045(22)00128-0. Epub 2022 Apr 4.), (J Immunother Cancer. 2022 Mar;10(3):e004316. doi: 10.1136/jitc-2021-004316.).
Author Response
We thank the referee for this advice and have incorporated additional discussion around PIV (line 477) and liquid biopsy (line 41) and included five of the useful references advised. Given that we are already above the recommended number of reverences, we do not think it appropriate to extend the discussion to include all of the points indicated in the further references identified.
Reviewer 2 Report
The article submitted by Wise and coll. is very interesting and well articulated. The experimental plan was suitably designed as well as the experiments were well performed and discussed. There are the following points which should be revised:
1. pag. 6, line 191: the explanation for the acronym PIN is reported, but PIN was used also in previous page. Therefore, the explanation for PIN should be given the first time it appears in the text (excluding the abstract). The same applies to another acronym on page 11, line 371, that is PKM2, the explanation of which is given only in the legend of Figure 5.
2. There is a confusion in the Supplementary Figures: at the end of the text, on page 13-14, as the supplementary figures should be 3, but in the file there are only 2 figures; moreover, the order of the legends is different from that reported in the file of Figures & Tables as well as along the Results section. Please, provide an adequate review of this part.
3. page 11, line 344, and pag 13, line 464: the Authors indicate the words: "Inflammatory response/Immunity"; however, in Supplementary Table 1, the words are "Innate immunity/Inflammatory response".
4. Page 6, line 184: "embroyonic" should be "embryonic".
Author Response
We thank the referee for pointing out the incorrect placing of abbreviations, which we have rectified. We have also provided a revised and corrected copy of the Supplementary figures and corrected the spelling of embryonic.
The lack of clarity regarding the name of immune gene expression signature stems from a change in identification of this cluster. The original assignation "Innate immunity/Inflammatory response" was deemed to be misleading as the cluster contains genes associated with both innate and adaptive responses. The list of relevant genes which show changes in our data also include genes which have been described to play roles selectively in both innate and adaptive immunity. We have clarified this point in the results (line 346).
Reviewer 3 Report
The manuscript by Wise and others identified a novel regulation of PTEN in prostate tumor development. The study identified that it is not the protein phosphatase activity but lipid phosphatase activity that regulates the tumor progression. Another conclusion of this work is that tumor development is driven by the total loss of PTEN function in the prostate in a manner that is not observed when PTEN function is partially lost. Also, the author identified some of the weaknesses of the study including the possibility of lacking protein phosphatase activity in immune cells and unexpected AR protein levels and nuclear localization.
Since this study indicates that there is no quantitative relationship between PTEN activity and tumor suppression having clinical relevance needs to be discussed in detail. Since the PTEN Y138L mice model only has a copy of the gene and can dephosphorylate the AKT to the extent of W/W also needs a supporting discussion or data. In addition, the influence on PIP3 or PIP2 phosphorylation by this mutation is required. Deriving the quantitative relationship between PTEN WT and PTEN Y138L levels with AKT phosphorylation and phosphorylation of the PI3K kinase products will elevate the significance and interest of the manuscript.
Author Response
Referee 3 identifies a challenge in this work: to understand the quantitative relationships between PTEN dose, PIP3, AKT activity and tumour suppression. We appreciate that our data do not fully address these uncertainties, but to clarify these points have added the following clarifying text in the discussion.
" The observation of PIN in prostates with a single copy of either wild-type Pten or PtenY138L but adenocarcinoma in Pten null prostates is consistent with previous observations of the dose dependency of tumour suppression in Pten mutant mice [29, 30]. It is also consistent with the observation that PTEN Y138L and the wild-type enzyme display very similar phosphatase activity against PIP3 and suppression of AKT in cell based assays ([46, 47] and Fig1A)."